# Learning from the Implementation of the Child Nutrition Program: A Mixed Methods Evaluation of Process

**DOI:** 10.3390/children9121965

**Published:** 2022-12-14

**Authors:** Emily DeLacey, Cally Tann, Tracey Smythe, Nora Groce, Michael Quiring, Elizabeth Allen, Maijargal Gombo, Merzel Demasu-ay, Batbayar Ochirbat, Marko Kerac

**Affiliations:** 1Department of Population Health, Faculty of Epidemiology and Population Health, London School of Hygiene & Tropical Medicine, University of London, Keppel Street, London WC1E 7HT, UK; 2Holt International, 250 Country Club Rd, Eugene, OR 97401, USA; 3Centre for Maternal, Adolescent, Reproductive & Child Health (MARCH), London School of Hygiene & Tropical Medicine, University of London, Keppel Street, London WC1E 7HT, UK; 4Department of Infectious Disease Epidemiology, Faculty of Epidemiology & Population Health, London School of Hygiene & Tropical Medicine, University of London, Keppel Street, London WC1E 7HT, UK; 5Neonatal Medicine, University College London Hospitals NHS Trust, 235 Euston Rd, London NW1 2BU, UK; 6MRC/UVRI & LSHTM Uganda Research Unit, London School of Hygiene & Tropical Medicine, University of London, Plot 51-59 Nakiwogo Road, Entebbe P.O. Box 49, Uganda; 7International Centre for Evidence in Disability, Department of Population Health, Faculty of Epidemiology and Population Health, London School of Hygiene & Tropical Medicine, University of London, Keppel Street, London WC1E 7HT, UK; 8Division of Physiotherapy, Department of Health and Rehabilitation Sciences, Stellenbosch University, Stellenbosch 7602, South Africa; 9UCL International Disability Research Centre, Department of Epidemiology and Healthcare, University College London, 1-19 Torrington Place, London WC1E 6BT, UK; 10Department of Medical Statistics, Faculty of Epidemiology and Population Health, London School of Hygiene & Tropical Medicine, University of London, Keppel Street, London WC1E 7HT, UK; 11Holt International Representative Office in Mongolia, Chinggis Avenue Mongol TV Tower 905, Sukhbaatar District 1st Khoroo, Ulaanbaatar 14251, Mongolia; 12Kaisahang Buhay Foundation, Inc., 1109 10th Ave, Quezon City 1109, Metro Manila, Philippines; 13Health Minister’s Office, Ministry of Health, Ulaanbaatar 15160, Mongolia

**Keywords:** child nutrition, caregivers, intervention programming, training of trainers, implementation, service and outcomes, child health, disease prevention

## Abstract

Nutrition and feeding interventions are important for children’s growth and development. Holt International’s Child Nutrition Program (CNP) is a child nutrition and feeding intervention. This study aims to describe and explore the implementation of CNP in Mongolia and the Philippines using mixed methods including qualitative and quantitative data analysis. The analysis framework was guided by the WHO’s Monitoring the Building Blocks of Health Systems. Key informant interviews (KIIs) were conducted, transcribed, translated and coded. Knowledge, Attitude and Practice Surveys (KAPS) and pre-/post-tests from routine program audit data were analyzed. Analysis of nutrition (Mongolia: 95% CI: 7.5-16.6 (*p* = < 0.0001), Philippines: 95% CI: 7.6-15.7 (*p*= < 0.0001)) and feeding (Mongolia: 95% CI: 11.7-23.9 (*p* = < 0.0001), Philippines: 95% CI: 6.6-16.9 (*p* = < 0.0001)) tests indicate improvement post-training in both countries. KAPS indicate changes in desired practices from pre-training to post-training. Thematic analysis of KIIs highlight essential components for program implementation and effectiveness, including strong leadership, buy-in, secure funding, reliable supply chains, training and adequate staffing. This evaluation of program implementation highlights successful strategies and challenges in implementing CNP to improve the health of children in Mongolia and the Philippines. Lessons learned from the implementation of CNP can inform growth of the program, scaling strategies and provide insights for similar interventions.

## 1. Introduction

Millions of children around the world continue to suffer from malnutrition for reasons including inadequate access to nutritious food, feeding practices and poor hygiene and sanitation [1,2,3,4]. Nearly half of the deaths among children younger than 5 years old have undernutrition as a primary factor [1,3]. Malnutrition predisposes children to long-term impairments such as disability, impaired cognition, non-communicable diseases and suboptimal performance at school [1,3]. How children are fed can be just as impactful as what they are being fed because both nutrition and feeding difficulties can heighten children’s malnutrition risk, especially for infants and children with disabilities [4,5,6,7]. Caregivers often need additional support to address children’s individual nutrition and feeding needs, especially if the needs and strategies to support the child are unfamiliar to the caregiver [8,9]. There is need for interventions which support children’s development and these programs need to be inclusive of children with disabilities and provide support to caregivers [6,8,9,10]. Evaluation of the implementation of such programs could provide insights into ways to enhance programs to better the outcomes for children and their caregivers [10,11,12].

Holt International is a 67-year-old child welfare non-profit working in 15 countries. In 2012, Holt identified that many of the children participating in its programs globally were at risk for malnutrition or experiencing malnutrition. Motivated to address this issue, Holt developed the Child Nutrition Program (CNP). The CNP works to address the critical nutrition, feeding, health and development needs of vulnerable children who are most at risk of malnutrition by providing training, resources and support for caregivers and sites providing care for children [13]. The program aims to improve individual and site level care practices. This program uses a Training of Trainers (ToT) approach in combination with formal assessment, monitoring and evaluation methods. Trained trainers who lead program implementation are considered CNP champions. The training enables caregivers and sites to conduct targeted interventions to address and prevent the causes of malnutrition, including undernutrition, overnutrition and micronutrient deficiencies for children, especially ages 0–5 and those with disabilities [14]. Training topics include child nutrition, feeding and positioning, hygiene and sanitation, growth monitoring, common illnesses, anemia screening, micronutrient deficiencies, disabilities and other development topics. The training is typically 5 days with a practicum and participants are selected by their sites and include staff from all positions. This program is implemented in community-based settings, foster care systems, health centers and institution-based care (IBC) [15]. After an initial site assessment, training is provided to site staff and caregivers followed by an evaluation and ongoing refreshment training and support from country level CNP teams. Training curriculum is standardized with some variations for context, such as maternal health and breastfeeding in community settings and formula feeding in IBC.

This study follows two retrospective analyses on the nutritional and feeding status of children who participate in the CNP [4,7]. This study aims to identify and explore key factors for program implementation through a mixed methods evaluation of process of the CNP in two countries—Mongolia and the Philippines.

### Objectives

Describe and assess the implementation of the CNP in Mongolia and the Philippines;Identify and describe the barriers, disruptions, enablers and solutions for implementation at a caregiver, site, country, multinational implementers and policy level; Explore key factors important for implementation and growth of the CNP.

## 2. Materials and Methods

### 2.1. Ethics Approval 

Ethical approval was obtained from the London School of Hygiene and Tropical Medicine (ref: 22865). The National Center of Public Health of Mongolia approved the research methodology/protocol and ethical approval was obtained from the Medical Ethics Control Committee of the Mongolian Ministry of Health (ref: 230). Ethical approval was obtained from the St. Cabrini Medical Center-Asian Eye Institute Ethics Review Committee (SCMC-AEI) Ethics Review Committee in the Philippines (ref: 2021-002). We have reported according to the TREND statement (Appendix A) [16]. A data use agreement was signed with Holt International for use of routinely collected de-identified program audit data. Both qualitative and quantitative data will be held indefinitely on Holt International’s server.

### 2.2. Study Design 

This study uses mixed methods to examine the implementation of a large multi-country nutrition and feeding intervention program. This study was designed by ED, MK, CT and TS and uses primary data collected by the principal investigator (PI), ED, and the CNP champions in Mongolia and the Philippines, in addition to secondary data from routine program audits of which all available data were used. The framework developed for this evaluation of process utilized a health systems approach guided by the WHO’s Monitoring the Building Blocks of Health Systems: A Handbook of Indicators and their Measurement Strategies [17,18,19]. 

### 2.3. Setting/Study Size

The CNP currently operates in eight countries at 68 sites serving over 7500 children. Countries include China, India, Mongolia, the Philippines, Ethiopia, Uganda, Vietnam and Haiti. The program operates in community-based settings, foster care systems, health care facilities and IBC. Mongolia began program implementation in 2016 and the Philippines in 2017. In the Philippines, there are seven sites consisting of five IBC and two foster care programs. In Mongolia, there are 13 sites including 4 health centers, 3 IBC, 5 schools/daycare sites and 1 facility with IBC, in-patient and outpatient services. Mongolia and the Philippines were selected for analysis because of logistics and data availability. 

### 2.4. Participants

This study utilized primary and secondary data. Participants in the primary data collection included key informants (KIs) who participated in key informant interviews (KIIs). KIs were selected using purposive sampling of one interviewee per site and per country program. The secondary data were collected during routine program audits. This data consisted of KAPS and nutrition and feeding pre-/post-training tests completed by staff at sites participating in the CNP as part of program implementation. These data components were analyzed using quantitative and qualitative methodologies and the findings were then combined to provide a broader synthesis of the CNP implementation.

### 2.5. Nutrition and Feeding Tests

The nutrition and feeding pre-/post-training tests were routine program audit data collected between 2016 and 2020 in Mongolia and the Philippines. The tests consist of questions on child nutrition and feeding information and practices which are covered in the CNP curriculum and training. The nutrition and feeding tests are repeated over time to identify areas of change and topics for future training for the program. The nutrition and feeding tests were analyzed from four collection time points: pre-training, post-training, six-months post-training/implementation and 1.5-years post-training/implementation, using descriptive statistics. Independent-samples *t*-tests were conducted comparing the unmatched pre-training and post-training tests (Table 1). 

### 2.6. Knowledge Attitude and Practice Surveys

The knowledge, attitude and practice surveys (KAPS) were collected during routine program audits prior to the site being trained in the CNP and after sites were trained between 2016 and 2020 in Mongolia and the Philippines. The KAPS were completed by staff of all levels at sites participating in the CNP. The surveys provide feedback to the program about participants knowledge, attitudes and practices in key programmatic areas. The surveys are routinely completed as part of program monitoring and evaluation systems and are used to track changes over time and inform trainers of key areas for training, as well as areas to support site implementation. The KAPS included respondent demographics and questions about nutrition, feeding, health, growth monitoring, disability, child development, hygiene and sanitation. There were a total of 25 questions; 11 knowledge-based questions, 5 attitude-based questions and 9 practice-based questions. As some of the questions reflected participants’ views and there were not “correct” answers, we summarized the KAPS using the program’s previously identified “desired” answers out of the total number of responses (Table 2, Table A1 and Table A2). Each question is color coded by its domain: knowledge questions in yellow, attitude questions in red and practice questions in green. The “desired” answer is specified in parentheses following the question (Table A1 and Table A2). 

### 2.7. Key Informant Interviews

The semi-structured KIIs were designed and pretested by the PI (ED) and the two CNP champions who are lead trained trainers and manage the program in the Philippines and Mongolia (Appendix A). The in-depth interviews consisted of open-ended questions on program implementation at the site or country level depending on interviewee (approximately 25–30 questions with prompts). Key informants were identified for participation by the CNP champions because they were site directors or lead staff who oversee the CNP implementation at their sites. The KIs had preexisting professional relationships with the CNP champions related to program participation. One informant from each site was identified for interview by the CNP champion in their respective countries. 

The CNP champions received interview training and practice prior to conducting interviews. All interviewees were provided and signed participant information and consent forms via DocuSign (DocuSign, Inc., San Francisco, CA, USA 2021) prior to the interviews. Interviews were conducted via password protected Zoom (Version 5.9.6, 2021). Interviews were transcribed, de-identified and then translated and shared with the PI (ED). The PI (ED) interviewed the two CNP champions on country level implementation. ED is the director of CNP, a lead trainer, has trained in all implementing countries and maintains relationships at the country level and site level. 

### 2.8. Statistical Analysis 

#### 2.8.1. Quantitative Methods

All quantitative statistical analyses were conducted using Stata (16.1, StataCorp LLC, College Station, TX, USA) and Microsoft Excel (2013). Independent-samples *t*-tests were used to compare the independent nutrition and feeding pre-training and post-training tests (Table 1 and Figure A1). A bar graph is used to notate the mean and confidence intervals for nutrition and feeding tests at pre-training, post-training, 6-month post-training/implementation and 1.5-years post training/implementation time points (Figure A1). 

Descriptive statistics were produced to summarize the independent KAPS. Respondent demographics and the frequency and percent of desired answers prior to training and after training are presented. A two-sided Fisher’s Exact test was used to assess whether there was any difference in the domains (knowledge, attitude and practice) from pre-training compared to post-training (Table 2). 

Demographic information of key informants who participated in the KIIs is presented. 

#### 2.8.2. Qualitative Methods

The qualitative data from semi-structured interviews were analyzed with descriptive coding for thematic content using NVivo 21 (released March 2020) and following the COREQ checklist [20]. The coding framework was developed to examine the KIIs with a health systems approach, which was guided by the WHO’s Monitoring the Building Blocks of Health Systems and its monitoring and evaluation of health systems strengthening [17]. The coding framework helped to process and systematically categorize qualitative data to identify themes and patterns in the interviews related to the process of implementation of the CNP. Defining and naming themes and grouping themes into categories were done by the PI with review by other co-authors. Codes were categorized and sub-codes were created. As key themes emerged, codes were consolidated. 

Following the WHO’s monitoring and evaluation of health systems strengthening inputs, processes, outputs, outcomes and impact were summarized by each of the six building blocks (Table A3). Additionally, a qualitative conceptual framework analysis of responses on barriers/disruptions and facilitators/solutions to implementation of the CNP was created by adapting the socioecological model produced by Rao et al., with areas identified by the analysis guided by the WHO’s health systems framework and building blocks (Figure 1) [21]. The information was further organized based on five levels: caregiver, site, country, multinational implementer and policy levels. For each level, facilitators/solutions and barriers/disruptions for implementing the CNP were identified. 

## 3. Results

In both Mongolia and the Philippines, the CNP was implemented at sites following standard program implementation starting with a formal assessment followed by a training and evaluation. After the evaluation, country level CNP staff provided ongoing support to sites and caregivers with additional training, resources and monitoring and evaluation systems. There were two main differences in context between Mongolia and the Philippines—the types of sites and who is trained. In Mongolia, many of the sites are health centers and daycares or schools for children with disabilities, which engage children’s caregivers, including mothers and fathers, and teachers or health center staff in training. In the Philippines, there are more IBC and foster care programs than in Mongolia, therefore training participants are primarily foster care parents or IBC staff. 

### 3.1. Nutrition and Feeding Tests

Analysis of the unmatched nutrition (Mongolia: 95% CI: 7.5–16.6 (*p* ≤ 0.0001), Philippines: 95% CI: 7.6–15.7 (*p* ≤ 0.0001)) and feeding (Mongolia: 95% CI: 11.7–23.9 (*p* ≤ 0.0001), Philippines: 95% CI: 6.6–16.9 (*p* ≤ 0.0001) tests suggest an improvement in knowledge and practices in both countries between the pre- and post-training (Table 1 and Figure A1). Additionally, there was a difference from the Philippines nutrition pre-training test to the 6-month post-training (95% CI: 10.9–21.8, *p* ≤ 0.0001) and 1.5-year post-training (95% CI: 9.9–21.5, *p* ≤ 0.0001). Differences at other test points may be due to chance, possibly related to changes in sample size or that the tests takers at different time points may not be the same. 

### 3.2. Knowledge Attitude and Practice Surveys

In total, 98 KAPS were analyzed. From the Philippines, there were 60 pre-training and 15 post-training KAPS from five sites. From Mongolia, there were 15 pre-training and 8 post-training KAPS collected from one site. Of the respondents from both countries, 96/98 (98%) were women and 65/98 (66%) had attended a university, graduate or professional school (Table A1 and Table A2). For Mongolia, the median age of respondents was at pre-training 42 years (IQR: 37–49 years) and post-training 43.5 years (IQR: 32–45.5 years). The majority of respondents in Mongolia had worked for more than 6 years (10/15, 66.7%) at pre-training and post-training the majority had worked for less than 6 years (6/8, 75%). In the Philippines, the median age of respondents was at pre-training 44 years (IQR: 32–50 years) and post-training 44 years (IQR: 37–49 years). The majority of respondents in the Philippines had worked for more than 6 years (35/60, 58.3%) at pre-training and post-training the majority had worked for less than 6 years (9/15, 60%).

There was an increase in desired answers from the practice domain and overall from the pre-training to post-training for both countries consistent with a positive change in implementation of practices. (Table 2). The Philippines saw an increase in desired answers from all three KAPS domains from pre-training to post-training. However, there were no statistically significant differences in KAPS outcomes after the training. This may be due to limitations of the samples such as change in sample size and the independences of the pre-training and post-training samples. Analysis of the KAPS in this study indicate that caregivers may need additional training and reinforcement, which is supported by the KIIs, which mention frequent training and retraining and integration of practices as essential to maintaining a high level of standardized program implementation. 

### 3.3. Key Informant Interviews

In Mongolia, 13 site directors or key staff and one CNP champion were invited to participate in KIIs, of which 10 staff from different sites and one CNP champion were interviewed. In the Philippines, eight individuals were invited to participate of which six agreed to participate. The six KIIs were conducted with individuals from five different sites and one interview was conducted with the CNP champion of the Philippines. All of the KIs were female. Of the types of sites, seven were from IBC facilities, six were community-based programs, such as schools for children with disabilities, one was a health center and a final site was a hospital that offers IBC, day-care/community services and inpatient/out-patient clinical services for children. KIs had participated in the CNP a median time of three years (IQR: 2–5 years) and their sites had participated for a median of four years (IQR: 2–5 years). The CNP champion in Mongolia has led the CNP for six years and the champion in the Philippines for four years. 

Analysis of the KIIs identified key barriers/ disruptions, facilitators and solutions to implementation (Figure 1). Key barriers identified included inadequate funding, insufficient supply chains, limited staffing and technology limitations. Key facilitators included partnerships, support, training, program buy-in from government officials and staff, secure supply chains, integration of practices and collaboration. The full analysis of the KIIs on the implementation is included in Table A3 [17]. The data were summarized and presented with guidance by the WHO’s health systems framework and building blocks. Additionally, key elements identified as essential for implementation at site, country and multinational implementers levels emphasize the need for clear communication, including memorandums of agreement or contracts with sites and partners (Table 2 and Table A3). Strong relationships and frequent training were also identified as essential elements at all levels. KIs recommended that for the CNP to be successful, sites and CNP program managers need to leverage sites’ commitment and success to advocate for growth through regional government leaders, as invested sites can share the value of the program and its impact. Sites sharing about the program could create traction for buy-in or interest from regional government leaders, translating into program growth at sites and buy-in to engage new sites in implementing the CNP. 

#### 3.3.1. Training and Behavior Change

Frequent training and retraining for all staff at the sites were the most frequently mentioned factors for sustained standardized program implementation over time. Training was reported as driving behavior change at a caregiver and site level. 


*“Since they were able to attend training and they know what its benefit is, I feel like the house parents can be encouraged to really do the practice,”—CNP Site Director*



*“Our employees’ passion and care for children, especially special needs [children with disabilities], have increased and changed positively. We learned to feed a child with swallowing and chewing difficulties. Children with disabilities, especially CP [Cerebral palsy], were fed with only very thin “liquidish” pureed food by bottles when they lay on their back. Now, we all use proper positioning as possible as their physical condition lets and feed them with proper food texture using cut out cups or maroon spoons adjusted with their abilities. We used to tell our children to sit quietly during meal times, but now we encourage them to communicate and interact with each other and our teachers improved their intention to interact with special needs kids,”—CNP Site Director*


#### 3.3.2. Technology and Health Screenings 

Technology, such as lack of access to the internet or computers, was identified as a key barrier to implementation. As part of implementing the CNP, sites are supported with supplies and access to the internet when needed and provided an electronic nutrition screening database. Use of the nutrition screening database was reported as an essential piece for implementation for both sites and country level implementers. The database allows sites to track and monitor children’s health and growth through analysis of health records. Participants frequently reported database use as a valuable tool in making other parts of their roles easier. Informants mentioned valuing the database due to its simplicity for monitoring and reporting of nutrition and health data which improves user experience and supports sites’ ability to easily report information to local government systems. 


*“We realize and see many positive changes in children’s health and development since implementing CNP at our site. We never had such [a] monitoring system and methods before. Now we can see the growth and nutrition progresses using CNP database. Children’s nutrition intakes and feeding quality were much improved and so their health condition became better,”—CNP Site Director*


#### 3.3.3. Program Understanding and Buy-in

Buy-in was also frequently noted as a key element to implementation. Buy-in and understanding of the program and its value to children is needed at all levels, including for caregivers, site staff, site leadership and other key stakeholders, such as local government. To achieve this, KIs suggested sharing of success from already existing programs, engagement of other stakeholders in training (i.e., parents or government officials or other organizations) and ensuring participatory training is part of the onboarding process for all of the new staff.


*“We try to organize some CNP trainings in extended scope and we intend to introduce the CNP to every one of the whole organization and we try to involve all level staff, including directors and also executive staff, also the children’s parents who have disabled children. We try to involve everyone who participates in taking care of the children. So I think it’s very important to make them understand of CNP,”—CNP Champion*



*“Maybe we can best achieve that [buy-in] by also, although we’ve done that already. We’ve sought help of the center head so it’s the sites’ leadership, so whenever we go to the regional director, we have center head with us, so that it’s not just KBF or Holt going to the regional office but also the center head. The sites and the sites’ center head goes along with us and shows that the site really has the need. So it goes two ways I think—so we connect the higher officials with the senior leadership and then we seek the support of the center head, so that we can have the center head share about the need and then she or he is able to go to the senior leadership and then say that, ‘Yeah this program is needed [at the site], and we really need it and that’s why we’re here to seek your support as well, so that whenever we need something we can ask you and can request anything from the senior leadership [government].’ Yeah, so I think that’s one way,”—CNP Champion*


#### 3.3.4. Alignment of Program with Site and Country Goals

KIs identified that the aims and objectives for nutrition programs or other services for children need to fit within sites and countries goals.


*“CNP complies with our organization’s medium-term strategic plan and the organization’s child protection policy by identifying barriers to learning, development and quality of life for every child with a disability that will have a positive impact on the child’s development and growth. It is also in line with the Mongolian Government policy for 2020-2024 program, ‘Vision-2050′—Mongolia’s long-term development policy, State Education Policy for 2014-2024 programs, the Convention on the Child Rights, the Child Protection Law and government resolutions,”—CNP Champion*


#### 3.3.5. Diversification of Funding

Funding was identified as necessary to implementation with a lack of diverse or secure funding being a key barrier. Often, funding or gifts-in-kind did not match site priorities. For example, sites received cookies instead of needed diapers, fruits or vegetables.


*“We found that it’s good to have partnership[s] with outside entities. We don’t want to be too dependent on one— because it’s very constricting. We’re boxed into the budget we receive,”—CNP Site Director*



*“When there are donors, it’s like—more on, not really for the kitchen or stocks, especially diapers, things like that. That’s the priority of the institution. Diapers, milk—things like that,”—CNP Site Director*


#### 3.3.6. Partnerships and Agreements

In both countries, implementation of this program worked top down from government relationships and bottom up from site level partnerships. KIIs reported a high value in signing agreements with clear expectations of all parties, including government agencies, suppliers and sites. Quality implementation and sustainability of the program correlated with government support, site partnerships and quality relationships with key stakeholders. Some of the sites have received recognition/awards from the government for their overall center quality, which included implementation of the CNP. This was reported to help to reinforce site commitment to the program.


*“We have reached not only the center head of the site, but also the regional directors so we conducted meetings with them and then we’ve made memorandum of agreement with them, though there’s like it’s not implemented right to the right for every word for word that’s in there but we have to be flexible, with what the site needs, but I think the partnership is there and in trusting each other to conduct this together and troubleshoot or whenever there’s like this needed help/there’s needed assistance, we can support each other in a way on how to make CNP doable for everyone,”—CNP Champion*


#### 3.3.7. Dissemination and Growth 

Dissemination and raising awareness about the program and its benefits was identified as essential for program growth. Sites that requested the program were noted to have high levels of implementation. These sites often referred new sites to the country level CNP staff to utilize the program based on the impact it has on children’s development at their site. KIs suggested engaging site directors from currently implementing sites into meetings with government officials, new potential sites and other stakeholders to share about program operation and impact. Additionally, the need for advocacy and awareness of the program was identified with suggestions to develop a focused media strategy and better share about research on the program with wider audiences. 


*“And I think one step for that, aside from the ongoing attempt to expand this to [new CNP sites], we plan also for gathering existing current champions and creating a best practices manual or anything that can be shared to anyone to see how CNP has been successful here. So I think my vision is something like that, so we can easily inform other people and other sites about CNP so that acceptance of the program can be easier,”—CNP Champion*



*“I think it will be very helpful to involving some of those Public Health National Center and also Health Ministry and Educational Ministry and Social Welfare Ministry people for their attention because you know, Mongolia has like straight line managing system, so those ministries are the most upper level supervising and managing and also developing strategy and policies for those sites, so I think their involvement would be helpful to scaling our program because CNP has lots of benefits for those vulnerable population,”—CNP Champion*



*“Maybe we can find someone who can somehow make nice about the CNP and really put CNP out there. Really make it popular somehow or make it really known to most people because, like when we think of businesses, when we think of important information we’ve been to like make it like trending or sensationalize… to put it out there, to really make it known in a way. Like maybe have someone who’s good at communication [or] publishing. And maybe for this research as well. If this research goes well and it finishes, then we can publish it further and then share with the scientific bodies, the experts and then show them [the value of the CNP],”—CNP Champion*


## 4. Discussion 

Exploration of the implementation of the CNP in Mongolia and the Philippines provides key insights that have the potential to increase the sustainability of the CNP interventions and improve implementation. The goal of this research was to be relevant and practical for implementers of CNP and potentially inform wider nutrition and feeding interventions. This research informs CNP staff, partners and similar programs on implementation strategies, as well as areas for future research.

Interviews with CNP champions and lead site staff provided key insights for implementation in the different contexts in which the CNP operates (Figure 1, Table 3 and Table A3). They identified many commonalities of barriers/disruptions regarding funding, staffing, dependence on donations, supply chains, COVID-19, reinforcement of site-wide practices and behavior change (Figure 1). In both countries, many KIs reported frequent training, integration of practices into site systems, incentives, strong local government relationships and strong oversight of implementation were facilitators/solutions to implementation at their sites. Securing diverse funding, strong partnerships, frequent communication, appropriate technology, routine monitoring and evaluation systems helped to mitigate disruptions related to staffing or leadership turnover, COVID-19, inflation and inadequate supply systems. Similarly to other research on implementation, we found the quality of implementation and sustainability of the program was related to strong leadership, frequent oversight, quality relationships, clear partnerships, training and government support, which are congruent with recommendations from the World Health Organization [11,22,23]. For the CNP, implementation of the program in different site types (community, foster-care, health centers and IBC) and country contexts highlights the adaptability of the program and the universal value of core nutrition and feeding training and education for caregivers [10]. KIs reported some of the success of CNP in their countries was due to the program fitting within the country’s goals for child nutrition and development, which made it easier to promote with local and regional governments [24,25]. 

Our findings show that successful sustained implementation of the CNP requires behavior change at both a caregiver level and a site-wide level. At both levels, behavior change was strongly linked to frequent training, hands-on practicums, clear roles and responsibilities, support, access to resources and adequate staffing. Similar to other programs, such as Ubuntu, Baby Ubuntu and Juntos, the CNP is structured to provide participatory training, resources and support to caregivers [12,26,27]. Training for caregivers can have a substantial impact on their behaviors, practices and feelings of support [5,8,10]. Participatory training, such as the CNP, can result in improved quality of life for caregivers and support them to improve practices and keep children healthy [10]. These findings are comparable to those reported by other programs, such as Ubuntu/Getting to Know Cerebral Palsy (CP), which found that with support and training, caregivers can have positive changes in their attitudes toward the children they care for and an improved understanding of children’s needs [8,10,12]. Similar to findings from Ubuntu, we found that caregivers can make significant gains in their knowledge and confidence in caring for children from participating in the CNP [8,10,12]. Building confidence in abilities for caregivers or staff at all levels was suggested by KIs as a key factor in implementation success [8,10,12,23]. Participating in the CNP also added value to caregivers’ personal lives in terms of use of practices at home, in the community and in their professional careers. As program managers move through the building blocks of the program health systems, they need to consider how behavior change methods can be integrated into their inputs and processes to achieve desired outcomes and improve impact [17,22,23].

Taking into consideration the results from the KIIs, KAPS, and nutrition and feeding pre-/post-training tests, behavior change and maintaining high quality implementation and integration takes ongoing support, frequent training and reinforcement (Table 2) [8]. Using insights from this research, currently implementing CNP sites and countries can review their implementation strategy for areas of reinforcement or improvement. This research indicates that for other similar nutrition and feeding programs, frequent training, building buy-in, support structures, involvement of key stakeholders, strong monitoring and evaluation need to be included in their program implementation. 

Next steps will be to use this information to explore different contexts and to investigate how best to scale up the CNP in countries where it currently operates, as well as future countries (Table 2). Involving other key stakeholders in the process, including children, caregivers, community members and government officials, will be essential [11]. The next steps in growing the CNP could look to using scaling frameworks, such as the WHO’s Nine Steps for Developing a Scaling-up Strategy, to determine how best to increase the range of impact of this program at both national and international levels [28]. Determining a scaling-up strategy could provide pertinent insights for expansion of the CNP and other similar programs.

### Strengths and Limitations

This study adds to limited evidence on implementation of nutrition and feeding intervention programs. We used a mixed-methods health systems approach to provide a more comprehensive evaluation of the process of implementing the CNP. The research practices were built upon strong existing relationships and took cultural protocols into consideration. Using mixed methods and analysis of different aspects of implementation enabled the research methods to complement each other to understand a complex implementation process more fully. This study included data and interviews from KIs in two countries of the eight where the CNP is implemented. These countries were selected because of data availability and logistics, but future research could look at implementation in all the countries were CNP is implemented. Conducting remote KIIs enabled this research to be efficient and less time consuming for our research team and interviewers although there is potential sampling and recruitment bias, as not all KIs were able to participate. Responder bias could also be present, as these KIs work at sites that benefit from support from Holt. Additionally, the principal investigator and some of the co-authors on this research are trainers or CNP champions who lead this program and whilst this can potentially introduce some influence, it also allows for enhanced analysis of the data because of a deep understanding of program operations and relationships with sites. Future research could consider using other methods such as use of control groups or independent evaluators to address the potential bias in this paper. 

As KAPS and nutrition and feeding pre-/post-training test samples were independent and there were smaller sample sizes at different time points, this could impact analysis of the results. These tests were conducted as part of routine program operations with unknown validity. Future research could further examine these tools. This research was also impacted by the COVID-19 pandemic because implementation of the CNP adapted as sites navigated through changes in public health restrictions. Other limitations included the fact that KIIs did not include other important stakeholders including children, caregivers, community members, families or government partners. 

## 5. Conclusions

The implementation of the CNP in Mongolia and the Philippines provides insights for implementation in other countries and for other similar nutrition and feeding interventions, in addition to areas for future research. With appropriate inputs, processes and implementation methods to address barriers and facilitators, programs such as the CNP could have the potential for substantial impact and growth. Strong partnerships and relationships with local government, secure funding, buy-in at all levels, adequate staffing, frequent training, support systems and adequate supply chains were identified as essential to implementation. As malnutrition continues to impact millions of children, programs that address the needs of caregivers and children, such as the CNP, should be prioritized. Applying scaling frameworks to future research on the CNP could provide additional information on how to scale-up programs to reach more children globally.

## Figures and Tables

**Figure 1 children-09-01965-f001:**
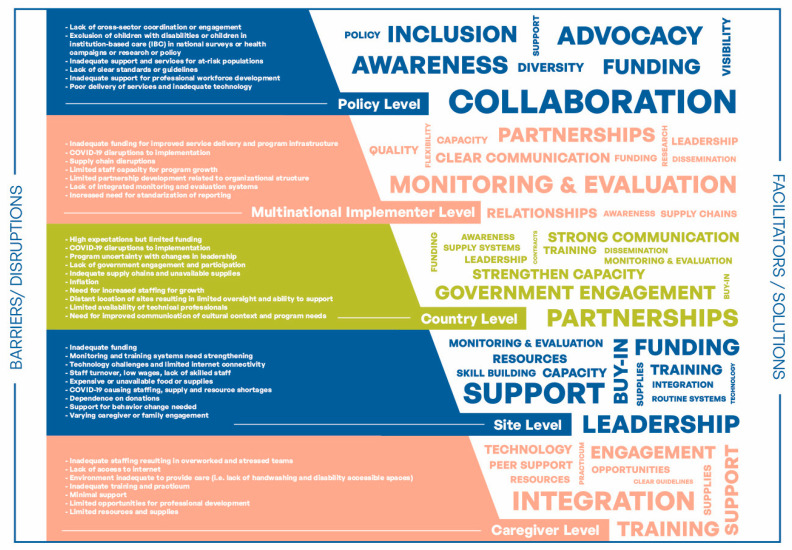
Synthesis of Barriers/Disruptions and Analysis of Facilitators/Solutions to the Implementation of CNP from the KIIs According to Levels of Conceptual Framework.

**Table 1 children-09-01965-t001:** Comparison of Unmatched Nutrition and Feeding Test Scores from Pre-training to Post-training, 6-month Post-training and 1.5-year Post-training using an Independent Samples *t*-test.

Mongolia														
Nutrition								Feeding						
	Summary Statistics				Independent Samples *t*-Test			Summary Statistics				Independent Samples *t*-Test		
	N	Mean	Median	Mean Difference (%)	df	*p*-value	95% CI	N	Mean	Median	Mean Difference (%)	df	*p*-value	95% CI
Pre-training (Reference)	45	70.8	70	REF	REF	REF	REF	39	62.9	66.7	REF	REF	REF	REF
Post-training	42	82.9	85	12	85	<0.0001	7.5–16.6	34	80.7	83	17.8	71	<0.0001	11.7–23.9
6 Month Post-training	12	65.3	66.7	5.5	55	0.151	−13.2–2.1	12	55.6	58.3	7.3	49	0.185	−18.3–3.6
1.5 Year Post-training	39	74.8	73.9	4	82	0.143	−1.4–9.2	39	65.2	66.7	2.3	76	0.505	−4.5–9.1
**Philippines**														
**Nutrition**								**Feeding**						
	**Summary Statistics**				**Independent Samples *t*-test**			**Summary Statistics**				**Independent Samples *t*-test**		
	N	Mean	Median	Mean Difference(%)	df	*p*-value	95% CI	N	Mean	Median	Mean Difference (%)	df	*p*-value	95% CI
Pre-training (Reference)	63	66.4	68.2	REF	REF	REF	REF	63	68.1	73.3	REF	REF	REF	REF
Post-training	58	78	80	11.7	119	<0.0001	7.6–15.7	57	79.9	86.7	11.8	118	<0.0001	6.6–16.9
6 Month Post-training	29	82.7	82.6	16.4	90	<0.0001	10.9–21.8	29	77.5	80	9.3	90	0.004	3.1–15.6
1.5 Year Post-training	19	82.1	84.8	15.7	80	<0.0001	9.9–21.5	20	77.3	83.3	9.2	81	0.019	1.5–16.8

**Table 2 children-09-01965-t002:** Change in Desired Answers of Knowledge, Attitude and Practice Surveys Between Pre-training and Post-training Using a Fisher’s Exact Test.

Mongolia			
Knowledge, Attitude and Practice Survey			
	Observations	Observations	
	Pre-training (n/N)	Post-training (n/N)	Two-sided Fisher’s Exact test
Knowledge	73.8% (107/145)	67.9% (57/84)	*p* = 0.749
Attitude	70.2% (40/57)	60% (24/40)	*p* = 0.742
Practice	58.6% (75/128)	81.9% (59/72)	*p* = 0.170
Overall	67.2% (222/330)	71.4% (140/196)	*p* = 0.673
**Philippines**			
**Knowledge, Attitude and Practice Survey**			
	Observations	Observations	
	Pre-training (n/N)	Post-training (n/N)	Two-sided Fisher’s Exact test
Knowledge	70.1% (499/712)	72.5% (116/160)	*p* = 0.839
Attitude	61.8% (170/275)	66.1% (43/65)	*p* = 0.826
Practice	69.9% (356/509)	81.5% (97/119)	*p* = 0.318
Overall	68.5% (1025/1496)	74.4% (256/ 344)	*p* = 0.380

**Table 3 children-09-01965-t003:** Summary of key elements needed for program implementation at the site level, the country level and the multinational implementers level identified from the KIIs.

Key Elements Needed for Implementation of the CNP
Multinational Implementers Level
Strong relationships with partners, country programs and other key stakeholders Secure and adequate funding in addition to identification of new funding or partnerships for growth opportunities Organizational buy-in Integration of research Strong and clear program communication Accountable and informative multi-level monitoring and evaluation systems Strategic plans for advocacy and awareness efforts
Country Level
Strong relationships with partners, local government, and other key stakeholders Clear memorandum of agreements with sites Secure and adequate funding Appropriate technology Identification of new opportunities for growth of program Frequent communication and refresher trainings for sites Access to strong in-country supply systems Strong implementation of monitoring and evaluation systems
Site Level
Access to reliable supply chains Frequent training and retraining for staff Buy-in from site leadership Adequate staffing and integration of program into staff onboarding Internet and technology resources are available Secure and adequate funding and diversification of funding Integration of program into site practices and workflow Clear guidance and support for staff Frequent review and oversight of sitewide behavior change and practices

## Data Availability

Some data is available upon request after an approval process from Holt International.

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
