# Peer review of "Learning from the Implementation of the Child Nutrition Program: A Mixed Methods Evaluation of Process"

_children, 2022, doi:10.3390/children9121965_

Round 1

Reviewer 1 Report

This manuscript describes and assesses the implementation of a Child Nutrition Program in Mongolia and the Philippines. Also it identifies and explores key factors for program implementation in order to the growth of the program, which can be very useful for implementation in other countries and for other nutrition and feeding interventions. The manuscript is well written, informative and I have no concerns.

The Abstract is clearly written, and in general the manuscript is well organized. Introduction provides a good explanation and is very clear, as well as the sections ‘Materials and Methods’ showing quantitative and qualitative methods matching with the objectives proposed. ‘Results’ and ‘Discussion’ are well described too. The authors have done a good job.

Author Response

Response to Reviewer 1

This manuscript describes and assesses the implementation of a Child Nutrition Program in Mongolia and the Philippines. Also it identifies and explores key factors for program implementation in order to the growth of the program, which can be very useful for implementation in other countries and for other nutrition and feeding interventions. The manuscript is well written, informative and I have no concerns.

The Abstract is clearly written, and in general the manuscript is well organized. Introduction provides a good explanation and is very clear, as well as the sections ‘Materials and Methods’ showing quantitative and qualitative methods matching with the objectives proposed. ‘Results’ and ‘Discussion’ are well described too. The authors have done a good job.

Response:

We appreciate your review of our paper and for highlighting the paper's clarity, organization, and full descriptions of all the sections. Thank you for your time spent reviewing this work. 

Reviewer 2 Report

Dear Authors:

Regarding the manuscript with title “Learning from the Implementation of the Child Nutrition Program: A Mixed Methods Evaluation of Process”, I only have some minor comments to address. This study is of high relevance as it adds well written and well organized information to limited evidence on implementation of nutrition and feeding intervention programs.

Comment 1:

Lines 28-29: Authors must change “Holt International’s Child Nutrition Program (CNP) is a nutrition and feeding intervention” by “Holt International’s Child Nutrition Program (CNP) is a child nutrition and feeding intervention program”.

Comment 2:

Lines 39-40: “Nutrition and feeding interventions are important for children’s growth and development “. This sentence should come in the beginning of the Abstract

Comment 3:

What is the difference that authors considered between nutrition and feeding?

Comment 4:

References should come before Appendix

Author Response

Response to Reviewer 2

Thank you for your time spent reviewing our paper. We appreciate your feedback that the paper was well written and organized. 

Comment 1:

Lines 28-29: Authors must change “Holt International’s Child Nutrition Program (CNP) is a nutrition and feeding intervention” by “Holt International’s Child Nutrition Program (CNP) is a child nutrition and feeding intervention program”.

Response:

Thank you for this suggestion, we have made this change. Line 29. 

Comment 2:

Lines 39-40: “Nutrition and feeding interventions are important for children’s growth and development “. This sentence should come in the beginning of the Abstract

Response:

We appreciate this suggestion and have made this change. Line 28.

Comment 3:

What is the difference that authors considered between nutrition and feeding?

Response:

To be clearer on the difference between feeding (method of eating) and nutrition (content of their diet) we have added a new sentence to the text. Line 55-56.

Reviewer 3 Report

The literature indicates obesity is an increasing problem in the countries within which you work. How does your program relate to cases of child obesity?

Please provide a rationale for selecting only Mongolia and the Philippines.

Please provide validity coefficients for the KAPS scales in comparable samples.

Please provide a rationale for why the semi-structured K11 interviewers were not blinded to program participation. Using CNP champions might induce social desirability responding?

Shouldn’t a non-program implementer have analyzed the interviews to avoid evaluation bias?

Why weren’t data collected on stakeholders blinded, instead of by the trainers/providers? Trainers/providers may provide positively biased perceptions (for a variety of reasons) which may not reflect actual participant outcomes.

Table 1: how can df be 85 etc. when there are 5 or less participants per test?

Table 1: the N: why are there specified N with no other data in the row?

Sample attrition from pre to post was high. Why?

Demographic characteristics of the caretakers and participant responders should be provided in the main report.

The lack of significant participant effects due to inadequate length of the program does not ring true. There are plenty of shorter programs that have achieved effects. The authors need to compare their findings to others in the literature.

If CNP had no participant effects, why is it being scaled up?

Table 2 indicates no statistically significant changes in participant outcomes, yet the authors conclude “…, programs such as CNP could have the potential for substantial impact …” This seems like a misrepresentation of the findings. The paper might be better focused on how to enhance the program to show improvements.

Table 1 in the Appendix presents responses for individual items. It is unlikely these are independent items, yet the sample is not large enough to conduct a factor analysis to identify internally consistent scales. Has any other published data used these items and conducted a factor analysis? Do the factors make sense in the context of Mongolia and the Philippines?

Same for Table 2.

Round 2

Reviewer 3 Report

The authors were reasonably responsive to this reviewer's comments.